# Yeasts Inoculation Effect on Bacterial Development in Carbonic Maceration Wines Elaboration

**DOI:** 10.3390/foods12142755

**Published:** 2023-07-20

**Authors:** Ana Rosa Gutiérrez, Pilar Santamaría, Lucía González-Arenzana, Patrocinio Garijo, Carmen Olarte, Susana Sanz

**Affiliations:** 1ICVV, Instituto de Ciencias de la Vid y el Vino (Universidad de La Rioja, Gobierno de La Rioja, CSIC), Finca La Grajera, Ctra. LO-20- salida 13, 26071 Logroño, Spain; 2Departamento de Agricultura y Alimentación, Universidad de La Rioja, 26006 Logroño, Spain

**Keywords:** wine, carbonic maceration, yeast inoculation, lactic bacteria, acetic bacteria

## Abstract

Carbonic maceration (CM) vinification is a very traditional method that allows saving energy without great equipment investment, obtaining high-quality wines. However, due to its particularities, CM winemaking implies a higher risk of microbial alteration. This work studies the evolution of bacterial population along carbonic maceration wines elaboration with and without yeast inoculation. In the same way, two strategies of yeast inoculation were studied: “pied de cuve” and Active Dry Yeasts (ADY) seed. For this purpose, three conditions were assayed: spontaneous fermentation (without inoculation), “pied de cuve” technology, and ADY inoculation. For each condition, two winemaking methods were compared: carbonic maceration and the standard method of destemming and crushing (DC). The bacterial evolution (lactic acid and acetic acid bacteria) was followed in different fermentation stages. Finally, the wines obtained were analysed (pH and volatile acidity). In the non-inoculated wines produced by CM, high development of the bacterial population was observed (counts of acetic acid bacteria around 4.3 log cfu/mL), and finished wines presented high values of volatile acidity (>1.5 g/L), which did not occur in the inoculated vinifications (counts of acetic acid bacteria around 1.5 log cfu/mL and 0.5 g/l of volatile acidity). Thus, the control of yeast population, as a “pied de cuve” as ADY seed, seems to be an effective tool to avoid bacterial alterations in CM vinifications.

## 1. Introduction

Carbonic maceration (CM) vinification is a traditional red wine-making technique that was used in different European wine regions such as La Rioja (Spain), *Beaujolais* in France or *Vino novello* in Italy, in the pre-phylloxera period, and has survived to the present day mainly in small wineries. 

In vinification by CM, the intact grape clusters, without destemming or crushing, are placed into tanks. Some grapes located in the lower zone of the tank are crushed by the weight and pressure produced by the ones above, releasing must to the bottom of the tank. This released must is fermented by yeasts, and the medium is progressively enriched in carbon dioxide. In these conditions, the whole grapes change from a respiratory metabolism to an anaerobic fermentative metabolism called intracellular fermentation (IF). The IF is carried out by grape enzymes inside the whole berries, triggering the production of alcohol, the degradation of malic acid, pectolytic and proteolytic phenomena, the formation of volatile compounds, and the diffusion of phenolic compounds from the skin to the pulp [1]. As a result of these processes, the whole grapes, at a certain moment, split open and release their juice into the tank, increasing the liquid phase, which continues to ferment due to the yeasts. Therefore, the IF of the grapes and the alcoholic fermentation (AF) of the must by yeasts occur simultaneously in the tank. After this first phase, drawing off is completed by racking a free-run wine, and the grapes that remain whole are pressed to release a higher-density press-wine. Then, a second phase begins when both wines—mixed or separate—complete their AF by the presence of yeast and malolactic fermentation (MLF) by lactic acid bacteria (LAB). 

This technique, which uses whole grapes without destemming or crushing, saves energy and investment in machinery by minimising processing equipment. In this vinification system, the release of calories during vinification is slower due to the fact that the fermentation is discontinuous, which reduces the consumption of frigories for temperature control during the process [2] This is due to the fact that the whole grapes that are broken down during the vatting process release a must rich in sugars, slowing down the fermentation process, so that the release of calories is slower and their release through the cap and the tanks is easier. In addition, the presence of stems also absorbs calories by preventing the must temperature from rising. In addition, the optimum temperature for intracellular fermentation is 30 °C, so there is no need for an exhaustive control of the temperature of the tanks. Because of this, this winemaking system has been maintained mainly in warm areas, and many small wineries continue to make this type of wine without external temperature control [3]. All these conditions would be an advantage in grape processing under the current climate change conditions [1].

Moreover, even today, small wineries still generate anaerobiosis in the tanks by fermenting the must from the broken grapes, making the external supply of CO_2_ unnecessary, which also means savings in the consumption of this gas and in its transport to the winery facilities.

All of the above means significant energy savings in wineries, without forgetting that this method of winemaking produces quality wines, different from those produced by the traditional system of destemming and crushing, which are in great demand by some consumers. 

However, due to its particularities, CM winemaking implies a higher risk of alterations due to the uncontrolled development of bacterial populations: lower levels of SO_2_, enrichment of must due to IF, oxygen trapped in the mass, etc. It is assumed that the wines produced by CM always have lightly higher volatile acidity values than those produced by the destemming and crushing (DC) method [4].

Controlling the development of the bacterial population during the winemaking process is essential for obtaining the correct wines. Bacteria, mainly lactic acid bacteria (LAB) and acetic acid bacteria (AAB), can cause defects in wines: increase in volatile acidity, undesirable aromas (geranium, butter, etc.), increase in the concentration of biogenic amines, appearance of ethyl carbamate, bitterness, smokiness, etc [5].

Lactic acid bacteria also play a positive role in winemaking by carrying out malolactic fermentation. This process, in which malic acid is transformed into lactic acid and carbon dioxide, usually takes place after alcoholic fermentation and is particularly desirable for red wines. Among the main lactic acid bacteria found in wines are: *Lactobacillus brevis*, *Lactobacillus higardii*, *Leuconostoc mesenteroides*, *Oenococcus oeni*, *Lactobacillus casei*, *Lactobacillus plantarum*, *Pediococcus damnosus* and *Pediococcus pentosaceus* [6]. Although malolactic fermentation is mainly produced by *Oenococcus oeni*, many of these bacteria are present during vinification and can be disruptive. Thus, heterofermentative bacteria (*L. mesenteroides*, *L. brevis*, *L. casei*, *L. hilgardii*, *L. plantarum*) can produce acetic acid from pentoses and produce biogenic amines (*L. hilgardii*, *P. parvulus*) or ethyl carbamate (*L. brevis*, *L. buchneri*) [6].

The population of lactic acid bacteria present in grapes is low. Normally, in the must and at the beginning of alcoholic fermentation, the population of lactic bacteria of some species increases, depending on the level of sulphite. During alcoholic fermentation, the population decreases and there is a selection in favour of the species more resistant to alcohol and pH. At the end of alcoholic fermentation, yeasts begin to decrease, the population of lactic bacteria, mainly *Oenococcus oeni*, increases, and malolactic fermentation begins [7].

Acetic acid bacteria are strictly aerobic, Gram-negative, non-sporulating, ellipsoidal, or round in shape. They are ubiquitous in nature and obtain energy mainly by the oxidation of sugars and ethanol to acetic acid [6]. In addition to the presence of O_2_, their growth can be influenced by other factors like pH and temperature. The optimum pH for the growth of acetic bacteria is between 5 and 6, so high pH favours their development in wine [8]. The optimum growth temperature of acetic bacteria is around 30 °C, although growths of acetic bacteria have been detected even at 10 °C, but much slower [9]. In oenology, the most common species are *Acetobacter aceti*, *Acetobacter pasterianus* and *Gluconobacter oxydans*. However, the presence of other species in wines from all over the world is continuously being studied. 

Recent works that describe physicochemical and especially microbiological characteristics of CM vinifications are scarce and contradictory. The contradictory results could be due to a wide range of causes, such as the grape variety, the grape ripeness level, the vintage, or the winemaking conditions [10,11,12].

In the studies carried out so far, the reinforcement of the yeast population in this type of elaboration is adequate in achieving adequate anaerobiosis and allows controlling the bacterial population that causes alterations [13]. The use of ADY as a fermentation starter is widely employed to secure the process and avoid sluggish or stuck fermentation [14]. However, traditional winemakers are keen to limit its use in order to reduce oenological inputs. The preparation of an indigenous winery-made fermentation starter from grapes called “pied de cuve” is popular, especially in artisanal and organic farming systems [15].

This work aims to study the bacterial population present in the CM vinification method under different yeast inoculation conditions and compare it with DC vinification to evaluate inoculation as a strategy for bacterial control.

## 2. Materials and Methods

### 2.1. Assays in the Experimental Winery

Six vinifications in triplicate (18 tanks) were carried out in 270 kg vats of Tempranillo grapes at the ICVV experimental winery. Three batches of 6 tanks were made, in which different types of inoculation were carried out: spontaneous fermentation (Batch Control), “pied de cuve” (Batch “Pied de cuve”), and inoculation with ADY (Batch ADY), respectively. In each batch, 3 tanks were vinified by the destemming and crushing (DC) method and 3 were vinified by the carbonic maceration (CM) method. 

The CM vinifications tanks were filled with 30 kg of crushed grapes, without destemming, to simulate the breakage that occurs when filling large tanks. Afterwards, 170 kg/tank of whole grapes were added. The density of the must and temperature were monitored daily in all the tanks. The free must was daily separated and softly pumped over the grapes to homogenise the tank. All the tanks were placed in a chamber at 30 °C to favour the appropriate conditions for IF. The tanks were devatted when the density of the liquid was 1.000, and two fractions were obtained: free liquid in the tank (FCM) and the liquid obtained by pressing the solid mass with a pneumatic press (PCM). The proportions obtained for each fraction in the CM vinification were approximately 30–35% (FCM) and 65–70% (PCM). Subsequently, in each fraction, alcoholic (AF) and malolactic fermentations (MLF) developed, and, at the end of both processes, the wines were sulphited with 40 mg/L of SO_2_.

The differences between the three batches were the origin of the yeasts which conducted the AF. Vinifications of Batch Control were carried out spontaneously without inoculation of yeasts. The anaerobiosis required for this winemaking system was achieved using the exogenous addition of CO_2_ gas during the first three days of vatting. Anaerobiosis of the “Pied de cuve” and ADY batches was achieved by active fermentation of the free-run must in the tank by inoculation. Batch “Pied de cuve” was carried out by inoculation with a “pied de cuve”. For the preparation of “pied de cuve”, three days before the harvest, 300 kg of grapes was harvested from the same vineyard. The grapes were crushed, destemmed, and pressed, and the must fermented spontaneously for 3 days at 25 °C. After this time, 25 L of this “pied de cuve” was added to the tanks before filling them with 30 kg of crushed grapes and 145 kg of whole grapes. The yeasts population of “pied de cuve” obtained was composed of *Saccharomyces cerevisiae* (54%), *Hanseniaspora uvarum* (42%), and *Metchnikowia pulcherrima* (4%).

In Batch ADY, inoculation was made with commercial ADY added at the beginning of vatting. The inoculum was composed of one *Saccharomyces cerevisiae* strain and three non-*Saccharomyces* yeasts (*Lachancea thermotolerans*, *Torulaspora delbrueckii* and *Metchnikowia pulcherrima*) at a dose of 25% each.

For the DC vinifications, grapes were destemmed and crushed, and the tanks were filled with the liquid and skins. The cap formed when AF started was punched down daily and no pumping-over was performed. Vinifications were conducted at 25 °C, and devatting was made when the density of the must was 1.000, and only one fraction was obtained (DC). The Control, “Pied de cuve”, and ADY batches vinified by DC were produced following the same inoculation guidelines as those explained in the case of CM.

MLF occurs spontaneously in all the tanks.

In the 9 final wines (three types of wines: DC, FCM, and PCM per three batches), the pH and volatile acidity were measured according to the official European Community methods [16]. The pH was determined with a pH meter (micropH 2001 CRISON), and the volatile acidity was determined by titration with NaOH 0.1N of the volatile acids separated from the wine with water vapour. 

### 2.2. Microbiological Analysis 

The microbial population present (bacteria and yeasts) was analysed in four of the winemaking process steps, as follows: 24 h after vatting (sampling 1, S1), tumultuous fermentation (sampling 2, S2) at the end of alcoholic fermentation (sampling 3, S3) and at the end of the MLF (sampling 4, S4). Serial dilutions of the must or wine samples were plated onto different culture media. 

Lactic acid bacteria (LAB) were measured by plating onto an MRS medium (52 g/L MRS broth, 20 g/L agar, 50 mg/L pymaricine and 0.1 g/plate biphenyl) and incubation at 30 °C under strictly anaerobic conditions for at least ten days. Acetic acid bacteria (AAB) were determined by seeding onto a Mann culture (25 g/L D-mannitol, 3 g/L peptone, 5 g/L yeast extract, 20 g/L agar, 3 U/mL penicillin, 50 mg/L pymaricine and 0.1 g/plate biphenyl) and incubation at 28 °C for 48 h. Total yeasts were measured by seeding onto a GYP culture medium (20 g/L glucose, 5 g/L yeast extract, 5 g/L peptone, 100 mg/L chloramphenicol and 0.1 g/plate biphenyl) and incubation at 28 °C for 48 h. When the incubation periods finished, the plates were examined, and colony-forming units per millilitre (cfu/mL) were counted in plates with 30–300 colonies. 

A maximum of 10 colonies per plate were isolated and processed for species identification using the MALDI-TOF system Matrix-Assisted Laser Desorption/Ionization-Time Of Flight) (Bruker Daltonik GmbH, Bremen, Germany), either directly or through the protein extraction sample preparation method according to manufacturer instructions. For the MALDI-TOF MS analysis, mass spectra were acquired using a Microflex Mass Spectrometer (Bruker Daltonik GmbH) with a nitrogen laser (λ = 337 nm) with flexControl software (Version 3.4; Bruker Daltonik GmbH).

### 2.3. Statistical Analysis of the Results

Analysis of variance (ANOVA) was carried out for all the analytical results determined in wines with the IBM^®^ SPSS^®^ Statistic version 26 (Armonk, New York, USA). Significant differences were established by using the Tukey post hoc test (*p* < 0.05–0.01). 

## 3. Results and Discussion

### 3.1. Characteristics (pH and Volatile Acidity) of Wines Obtained

The pH values of the 9 wines obtained ranged from 3.6 to 3.9. There were no significant differences between the three wines in each batch. However, the mean pH values showed slight differences between batches. Thus, the wines from Batch Control and “Pied de cuve”, with mean values of 3.8, had a slightly higher pH than the wines from Batch ADY. The wines from this batch, with a pH of 3.6, were slightly more acidic, which was probably due to the different yeasts present in the inoculum used, especially the *Lachancea* genus [17].

However, as shown in Figure 1, large differences were observed in the volatile acidity of the CM wines from Batch Control compared to the rest. Thus, FCM and PCM wines from this batch had a volatile acidity of 4.02 and 5.14 g/L, respectively, while in the rest of the wines, this parameter did not exceed 0.5 g/L. Values higher than 1.5 g/L are considered organoleptically inadequate (“acetic piqure”) [18].

Several oenological microorganisms can increase volatile acidity [19,20,21], and it is well known that, to avoid increasing the volatile acidity in wine, it is necessary to control the microorganisms present at all stages of the winemaking process. The yeast *Saccharomyces cerevisiae* generates small amounts of acetic acid during alcoholic fermentation [22], but it is non-*Saccharomyces* yeasts (*Hanseniaspora guilliermondii*, *Kloeckera apiculate*) that are the main producers [23]. Heterofermentative lactic acid bacteria (*Oenococcus oeni*, *Pediococcus* spp., *Lactobacillus* spp.) present in wine also slightly increase volatile acidity during malolactic fermentation and wine preservation under the right conditions [5]. However, AAB is the main producer of acetic acid in wine, but they are not usually common in wine. Certain factors favour the development of acetic acid bacteria, such as the presence of residual sugars and, above all, oxygen, which is the limiting factor for the growth of these micro-organisms. The absence of oxygen does not eliminate bacteria, but it does prevent their growth and slow down their metabolism [24].

### 3.2. Development of Yeasts

Figure 2 shows yeast evolution in the different batches and sampling times.

Among the three batches studied, no significant differences were found in the yeast populations throughout the vinification, although in Batch Control, the populations were slightly lower. A similar evolution of the populations within each batch was also observed in the different samplings carried out. In all cases, the residual yeast population in the finished wines (S4) was higher in the CM wines.

The differences observed in the volatile acidity of the wines from the three batches could be due to the different types of yeasts present, especially non-*Saccharomyces* yeasts (Figure 3), or the presence of other microorganisms (LAB, AAB).

The evolution of the yeast population in the winemaking process is key to establishing a correct level of anaerobiosis that directs the fermentation process, hindering the development of oxidative microbial activities that could be detrimental to the quality of the final wine [25]. This aspect is especially important in CM vinifications, in which, due to the special vinification process, significant amounts of oxygen are trapped in the grape mass.

Figure 3 shows the distribution of *Saccharomyces* and non-*Saccharomyces* yeasts in the different samples taken during the vinification of the three batches. In Batch Control, at the start of vatting, 100% of the yeasts were non-*Saccharomyces*, which could indicate a delay in establishing anaerobiosis in CM production. Although *Saccharomyces cerevisiae* eventually prevailed, this delay in establishing anaerobiosis could be decisive, as it would favour the development of LAB and AAB, and it would justify the high values of volatile acidity found in the CM wines of this batch. Also, in the inoculated batches (Batches “Pied de cuve” and ADY), non-*Saccharomyces* yeasts had an important presence in S1 (50 and 85% in DC wines and 20 and 90% in CM in Batches “Pied de cuve” and ADY) and were present until the end of the AF. However, the inocula used in these batches showed higher fermentative activity compared to the yeasts presented in the vinifications of Batch Control. In fact, in Batch “Pied de cuve”, the yeasts present in the inoculated “pied de cuve” were mostly *S. cerevisiae* (54%) together with *Hanseniaspora uvarum* (42%) and *Metchnikowia pulcherrima*. In Batch ADY, although the percentage of *S. cerevisiae* was 25%, the non-*Saccharomyces* yeasts included in the commercial inoculum used have been widely described as active fermenting microorganisms, mainly at the beginning of vatting [26].

### 3.3. Development of Lactic Acid Bacteria

Lactic acid bacteria can be detrimental and beneficial to the quality of the wine [27]. Their performance in wine is related to the specific species and strain genetics but also to many other factors, including environmental conditions, microbial interactions, and, mainly, the moment of winemaking in which each microorganism intervenes [28,29,30].

Figure 4 shows the evolution of the LAB population for the three batches.

In all cases, higher populations were observed in CM than in DC. This may be attributed to the winemaking conditions, especially to the release of N_2_ caused by the IF, which passes into the must as the grapes are broken down. This N_2_ enrichment of the must would favour the development of LAB in the rest of the stages. In S4, when the sample was taken after MLF, the increase in LAB detected is logical.

The greatest differences in LAB counts were found during the development of AF in Batch Control, detecting a high presence of LAB in the CM wines compared to those produced by DC. Thus, while LAB counts in DC in samples S2 and S3 were 3.26 and 2.58 log cfu/mL, respectively, in the CM elaborations in these same samples, they reached higher values (between 6 and 7 log cfu/mL). Their high presence in a medium still rich in sugars would explain the increase in volatile acidity in these wines, which is attributable to “lactic piqure” caused by lactic acid bacteria.

However, in batches “Pied de cuve” and ADY during AF, the differences in LAB populations between wines made by the DC and CM methods were not as pronounced: LAB counts in DC were 1.67 and 2.11 log cfu/mL in Batch “Pied de cuve” and 2.22 and 2.7 log cfu/mL in Batch ADY in the S2 and S3 samplings, respectively. In these samplings, in the CM vinifications, LAB populations did not exceed 5 log cfu/mL in any case. It can be deduced that inoculation controls the development of LAB at stages where their presence in high populations may cause disturbances.

Figure 5 shows the species distribution of LAB isolated at the different sampling times in the wines of the three batches studied.

It can be observed that in Batch Control, there was a higher participation of *Oenococcus oeni* in CM winemaking. Due to its high tolerance for low pH, high ethanol concentrations, and scarcity of nutrients, *O. oeni* is the main LAB of choice in winemaking. In CM wines of Batch Control, *O. oeni* had a notable presence in all stages of winemaking. Although it was only the majority species in the finished wines, the high LAB populations reached during winemaking and the heterofermentative character of this species posed a high risk of “acetic and lactic piqure”, as reflected in the high volatile acidity values found in the wines of this batch made by CM. In this batch, in the wines produced by DC, *O. oeni* was only detected in the S4 sampling, leading to a correct MLF that resulted in a correct wine concerning its volatile acidity [31].

In Batch “Pied de cuve”, a scarce presence of *O. oeni* was detected during AF in both DC and CM wines. This species was only the majority in S4, leading the MLF. In the wines of this batch, *Lactiplantibacillus plantarum* predominates. Since in these wines, the amount of LAB is lower and, moreover, since *L. plantarum* is homofermentative for hexoses, it does not produce volatile acidity through sugar metabolism. However, *Lactiplantibacillus strains*, with their fast consumption of malic acid (up to 3 g/L in 2–4 days) and the suppression of the activity of other spontaneous LAB populations, were able to conduct the MLF [32,33].

The appearance of *Pediococcus pentosaceus* in some samples does not seem to have had a major impact, as it is a purely homofermentative bacterium. The genus *Pediococcus* is generally considered a spoilage microorganism in wines [34]. *P. damnosus*, *P. inopinatus*, *P. parvulus*, and *P. pentosaceus* have been reported to produce excessive diacetyl, exopolysaccharides, biogenic amines, acrolein, and, more generally, off-odours, flavours, and textures, thus contributing detrimentally to wine quality [34]. However, recent findings have shown that the presence of *Pediococcus* species in wine does not always lead to spoilage and that some species and strains within this genus may contribute positively to wine aroma [34,35].

In Batch ADY, *O. oeni* dominated in all stages, not only in CM but also in DC, although in a lower population than in Batch C. Moreover, as the AF was controlled by the inoculation, the risk of attack on sugars and the appearance of “acetic and lactic piqure” is lower.

The differences in the distribution of *O. oeni* found in Batch “Pied de cuve” and ADY can be attributed to the type of inoculation practised. While in Batch “Pied de cuve”, the inoculation adds a diversity of micro-organisms (yeasts and bacteria) and species, in Batch ADY, the inoculation produces a rapid onset of AF and early alcohol accumulation, which *O. oeni* resists more than other species.

### 3.4. Development of Acetic Acid Bacteria

Figure 6 shows the evolution of the AAB population for the three batches.

As with LAB, the highest AAB populations were found in the CM wines from Batch Control, with counts of 5.15 and 3.53 log cfu/mL in the FCM and PCM finished wines, respectively. These high populations would also be related to the occurrence of the high volatile acidity values detected in the CM wines of this batch (both FCM and PCM). This would indicate, taking into account the data obtained in the LAB study, that both “lactic and acetic piqure” were produced in these wines.

However, yeast inoculation in Batches “Pied de cuve” and ADY was effective in controlling the AAB population in both DC and CM wines. All wines from these batches had AAB populations below 2.1 log cfu/mL and maintained volatile acidity levels within the normal range.

As in LAB, the AAB species present also could influence the final wines. However, it is necessary to comment that some isolates were not acetic acid bacteria but environmental microorganisms, which also grow in the Mannitol medium, which is something that we had detected previously [36].

Figure 7 shows the species distribution of AAB.

In the DC vinifications of Batch Control, the low populations of bacteria isolated on the MAN plates were not identified as AAB from the S3 sampling.

The absence of high populations of this type of bacteria during the vinification processes and in the finished wines prevents the occurrence of “acetic piqure” [8].

Acetic bacteria were also not found in the isolates of the finished DC wines (sampling S4) from Batch “Pied de cuve”. On the other hand, the AAB bacteria isolated from the DC wines of Batch ADY corresponded entirely to *Gluconobacter oxydans*. This bacterium, in the populations found (around 2 log cfu/mL), did not significantly increase the volatile acidity of these wines.

In the final wines (S4) produced by CM from Batch Control, the high bacterial counts developed on MAN agar corresponded to 50 and 40% non-acetic bacteria for FCM and PCM, respectively. The AAB identified belonged mostly to the genus *Gluconobacter*, while *Acetobacter pasteurianus* represented only 20% of the isolates in FCM wine and 10% in PCM wine. On the other hand, in the final wines (S4) elaborated by CM from the batches in which yeast inoculation was performed (Batches “Pied de cuve” and ADY), the few isolates found were mainly identified as *Acetobacter pasteurianus*, accounting for 100% in the wines of Batch ADY and only in the FCM wine of Batch “Pied de cuve” was this bacterium found in 57% together with *Gluconobacter oxydans*.

The differences in counts (probably due to the different kinetics of anaerobiosis establishment) and the different metabolic pathways used by the *Acetobacter* and *Gluconobacter* genera would explain the absence of an “acetic piqure” in the DC wines and the CM wines of the “Pied de cuve” and ADY batches. The genus *Gluconobacter* was described for its intense capacity to oxidize glucose to gluconic acid rather than ethanol to acetic acid and no oxidation of acetate, which were different in these respects from strains of the genus *Acetobacter* [37]. *Acetobacter aceti* was not found in any of the finished wines. *A. aceti* normally grows in finished wines because it metabolises alcohol and is closely related to the “acetic piqure” of wines. The occasional identification of this bacterium at other sampling times in some vinifications did not have a negative impact on the final wines.

The greater microbial variability found in the uninoculated Batch Control wines would help to explain their higher volatile acidity values, especially in the wines made by CM.

Therefore, although there was a higher population of LAB and AAB in the CM wines, it seems that the deterioration of the Batch Control wines was mainly due to LAB activity during the vatting phase when appreciable sugar concentrations were still present.

Although the increased presence of lactic acid bacteria during CM vinification was described by Flanzy et al. [38], the detection of high populations of acetic acid bacteria is more surprising considering that the CM process is carried out under anaerobic conditions. However, there is no previous work comparing bacterial populations and species distribution in MC and DC vinification nor of the main species present in each vinification system. Therefore, further studies are needed to verify the differences detected in this work, which would justify the higher risk of alteration of these wines.

Although the yeast inoculation in cellar tanks is a widely used control practice in winemaking, most CM wines are produced in artisanal wineries, where yeast inoculation is not common. However, this work shows that yeast inoculation at the beginning of vatting in CM vinification is a very useful tool for AF control. This simple tool makes it possible to control winemaking and avoid deviations, both in the “pied de cuve” format and by using active dry yeasts.

However, the “pied de cuve” strategy is still empirical in wineries, because it involves the contribution of an unknown microbial population, both yeasts and bacteria. Therefore, in addition to being a more difficult technique to perform properly in artisanal wineries, it implies less control of the microbiota added to the tanks. On the other hand, ADY inoculation implies better knowledge of the micro-organisms added, avoiding the participation of non-fermenting yeasts capable of increasing volatile acidity, as is the case with *Hanseniaspora uvarum*. If acidifying yeasts, such as *Lachancea thermotolerans*, are also included in the inoculum used, the pH and, therefore, the development of unwanted bacteria is reduced [39].

## 4. Conclusions

This work shows the risk of “acetic and lactic piqure” caused by alterations in the bacterial development inherent to the CM method of winemaking and the convenience of conducting bacterial evolution along this winemaking method. This is the first published work to describe qualitative differences in the distribution of lactic and acetic acid bacteria between this method of winemaking and the reference method of destemming and crushing. Moreover, yeast inoculation, by adding a “pied de cuve” or ADY, seems to be an effective tool, which would allow greater control of this traditional winemaking process by avoiding the use of exogenous CO_2_ and minimising the risk of spoilage.

## Figures and Tables

**Figure 1 foods-12-02755-f001:**
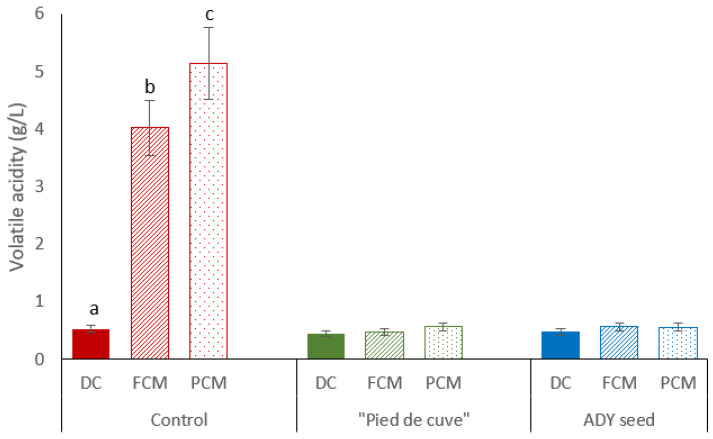
Volatile acidity of the nine final wines of the three batches elaborated (Control: without inoculation; “Pied de cuve”: with addition of a “pied de cuve”; ADY: with Active Dry Yeast seed). DC (solid colours): destemming and crushing wines; FCM (diagonal stripes): Free wines obtained by carbonic maceration vinification; PCM (dotted): Press wines obtained by carbonic maceration vinification. The error bars represent the standard error of the mean. Different letters on the columns at the same batch mean significant differences between wines (*p* ≤ 0.05).

**Figure 2 foods-12-02755-f002:**
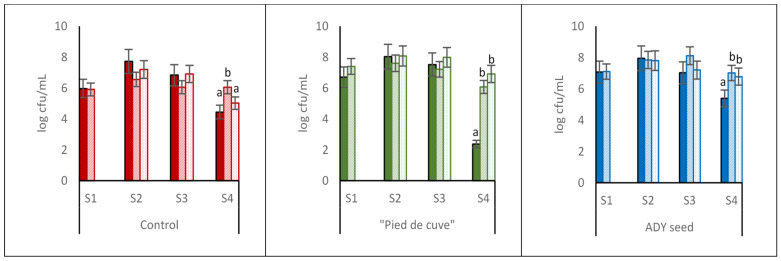
Counts of yeasts in the three batches analysed (Control: without inoculation; “Pied de cuve”: with addition of a “pied de cuve”; ADY: with Active Dry Yeast seed) at different moments of winemaking (S1: 24 h; S2: tumultuous AF; S3: end of AF; S4: end of MLF). DC (solid colours): destemming and crushing wines; FCM (diagonal stripes): Free wines obtained by carbonic maceration vinification; PCM (dotted): Press wines obtained by carbonic maceration vinification. The error bars represent the standard error of the mean. Different letters on the columns at the same sampling moment mean significant differences between wines (*p* ≤ 0.05).

**Figure 3 foods-12-02755-f003:**
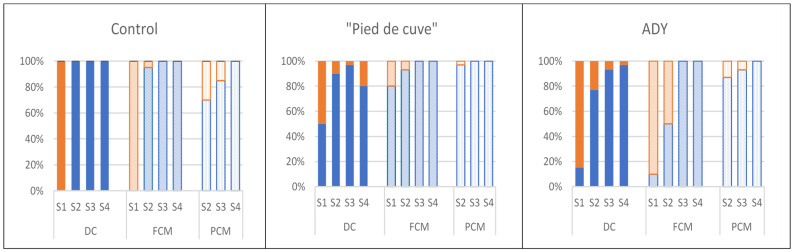
*Saccharomyces* (
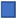
) and Non-*Saccharomyces* (
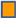
) distribution in the three Batches analysed. (Control: without inoculation; “Pied de cuve”: with addition of a “pied de cuve”; ADY: with Active Dry Yeast seed) at different moments of winemaking (S1: 24 h; S2: tumultuous AF; S3: end of AF; S4: end of MLF). DC (solid colours): destemming and crushing wines; FCM (diagonal stripes): Free wines obtained by carbonic maceration vinification; PCM (dotted): Press wines obtained by carbonic maceration vinification.

**Figure 4 foods-12-02755-f004:**
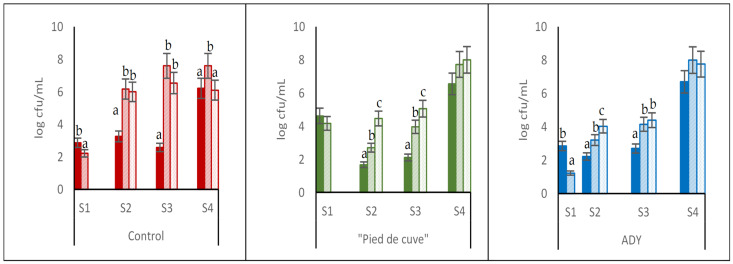
Counts of lactic acid bacteria in the three batches analysed (Control: without inoculation; “Pied de cuve”: with addition of a “pied de cuve”; ADY: with Active Dry Yeast seed) at different moments of winemaking (S1: 24 h; S2: tumultuous AF; S3: end of AF; S4: end of MLF). DC (solid colours): destemming and crushing wines; FCM (diagonal stripes): Free wines obtained by carbonic maceration vinification; PCM (dotted): Press wines obtained by carbonic maceration vinification. The error bars represent the standard error of the mean. Different letters on the columns at the same sampling moment mean significant differences between wines (*p* ≤ 0.05).

**Figure 5 foods-12-02755-f005:**
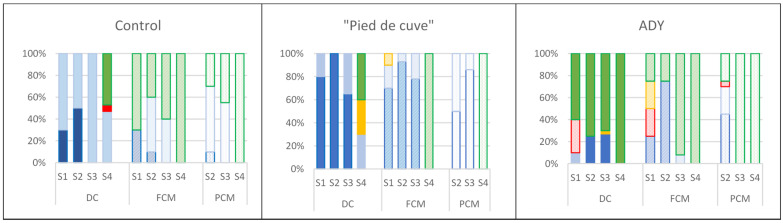
Species distribution of lactic acid bacteria in the three batches analysed (Control: without inoculation; “Pied de cuve”: with addition of a “pied de cuve”; ADY: with Active Dry Yeast seed) at different moments of winemaking (S1: 24 h; S2: tumultuous AF; S3: end of AF; S4: end of MLF). DC (solid colours): destemming and crushing wines; FCM (diagonal stripes): Free wines obtained by carbonic maceration vinification; PCM (dotted): Press wines obtained by carbonic maceration vinification. 
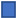

*Lactiplantibacillus plantarum*; 
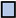

*Lactobacillus* sp.; 
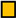

*Pediococcus pentosaceus*; 
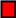

*Leuconostoc* sp.; 
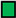

*Oenococcus oeni*.

**Figure 6 foods-12-02755-f006:**
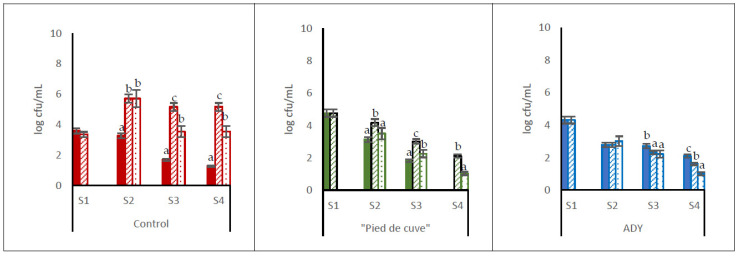
Counts of acetic acid bacteria in the three batches analysed (Control: without inoculation; “Pied de cuve”: with addition of a “pied de cuve”; ADY: with Active Dry Yeast seed) at different moments of winemaking (S1: 24 h; S2: tumultuous AF; S3: end of AF; S4: end of MLF). DC (solid colours): destemming and crushing wines; FCM (diagonal stripes): Free wines obtained by carbonic maceration vinification; PCM (dotted): Press wines obtained by carbonic maceration vinification. The error bars represent the standard error of the mean. Different letters on the columns at the same sampling moment mean significant differences between wines (*p* ≤ 0.05).

**Figure 7 foods-12-02755-f007:**
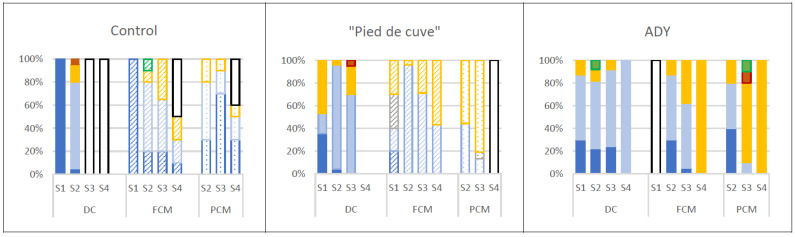
Species distribution of acetic acid bacteria in the three batches analysed (Control: without inoculation; “Pied de cuve”: with addition of a “pied de cuve”; ADY: with Active Dry Yeast seed) at different moments of winemaking (S1: 24 h; S2: tumultuous AF; S3: end of AF; S4: end of MLF). DC (solid colours): destemming and crushing wines; FCM (diagonal stripes): Free wines obtained by carbonic maceration vinification; PCM (dotted): Press wines obtained by carbonic maceration vinification. 
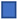

*Gluconobacter cerinus*; 
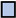

*Gluconobacter oxydans.*; 
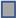

*Gluconobacter* sp.; 
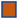

*Acetobacter aceti*; 
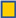

*Acetobacter pasteurianus*; 
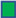

*Gluconoacetobacter intermedius*; 
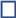

*Non acetic bacteria*.

## Data Availability

The datasets generated for this study are available on request to the corresponding author.

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
