# Peer review of "Yeasts Inoculation Effect on Bacterial Development in Carbonic Maceration Wines Elaboration"

_foods, 2023, doi:10.3390/foods12142755_

Round 1

Reviewer 1 Report

The work is average but may be improved by the inclusion of the following suggestions.

1.      English should be improved throughout the manuscript.

2.      Quantitative information should be provided in the abstract.

3.      The concussion should be concise and to the point indicating the application of the work.

4.      The novelty of the work should be established.

5.  Please provide error graphs in the figure; where are required.

6.      Please improve the quality of the Figures.

7.      Please compare your results with previous studies and mention clearly how your work is important in comparison to already been reported.

no

Author Response

The work is average but may be improved by the inclusion of the following suggestions.

  1. English should be improved throughout the manuscript.

English was reviewed and corrected by a translator as you can find in the acknowledgements. Moreover, the final and corrected version of the manuscript have been also carefully reviewed.

  1. Quantitative information should be provided in the abstract.

We have included quantitative information in the abstract.

  1. The conclusion should be concise and to the point indicating the application of the work.

We have modified the conclusions according to your suggestions.

  1. The novelty of the work should be established.

Some sentences describing the novelty of the work has been included in the conclusions section.

  1. Please provide error graphs in the figure; where are required.

The error bars have been included in the Figures 1, 2, 4, and 6. In contrast, in figures 3, 5 and 7 the error bars are not applicable since the data shown are expressed in percentage and standard deviation is not adequate for this statistic.

  1. Please improve the quality of the Figures.

The Figures have been improved. We hope that the modifications incorporated in the figures will make their interpretation easier.

  1. Please compare your results with previous studies and mention clearly how your work is important in comparison to already been reported.

In the revision of the literature at the beginning of a R&I project aimed to characterize the carbonic maceration winemaking any recent work dealing with the microbiology of this winemaking system was found. The only research papers published on the subject in the last 5 years have been carried out by us and they are all cited in this paper.

Reviewer 2 Report

Your manuscript is very interesting and the research you undertook is worthy of investigation.

I feel that you should enhance your introductory section by adding relevant references. For instance, you give a very interesting description about carbonic maceration (which is the topic of your MS), however, from ln 41 through to ln 70 you don't use a single reference. The very detailed description you are giving in lns 50-62 requires multiple references.

While CM is one of the topics of your MS, the use of various inoculums is the other topic of your MS. I believe that you are remiss in not covering the notions of both spontaneous fermentation and "pied de cuve" in your introduction.

ln 153, remove the word "the" in "In Batch S ..."

Ln 165. Rather than stating that pH and volatile acidity were measured according to ... - provide a simple description. Not everyone has access to those official methods.

Figure 1. rather than using "batch C", "Batch P", etc as descriptions, name the actual method. Furthermore, Use different colours for the different inoculums. And, give a description of the figure in the caption (what do the letters mean on the error bars, reiterate the abbreviations, make statement about reproducibility, etc.).

Ditto for Figure 2, which I find very confusing.

Ditto for Figures 3 and 4 and 5 and 6 and 7. Overall, tidy up the figure and captions.

Consider whether the data from some of your graphs are better presented in a table.

Try not to refer to "Batch S", "Batch P" or "Batch C". I know that is what you might have called them during the experiments, but - keep it simple, call them what they are.

Author Response

Your manuscript is very interesting and the research you undertook is worthy of investigation.

Thank you very much for your valuable comment.

I feel that you should enhance your introductory section by adding relevant references. For instance, you give a very interesting description about carbonic maceration (which is the topic of your MS), however, from ln 41 through to ln 70 you don't use a single reference. The very detailed description you are giving in lns 50-62 requires multiple references.

Since Flanzy described the CM winemaking process in the middle of the last century, any work that examines in depth the technology of this winemaking system has been published. The information included in lines 40-70 is derived from the author's description in the aforementioned work. In order to support what is indicated in the text, we have quoted this work in the paragraph.  We have also included in this section the book written by Hidalgo Togores where the CM vinification process is explained from the point of view of a Spanish oenologist.

While CM is one of the topics of your MS, the use of various inoculums is the other topic of your MS. I believe that you are remiss in not covering the notions of both spontaneous fermentation and "pied de cuve" in your introduction.

A paragraph with information on inoculation procedures and supporting citations has been included in the manuscript.

ln 153, remove the word "the" in "In Batch S ..."

The word has been removed.

Ln 165. Rather than stating that pH and volatile acidity were measured according to ... - provide a simple description. Not everyone has access to those official methods.

A brief description of the methods of analysis of pH and volatile acidity has been included in the manuscript

Figure 1. rather than using "batch C", "Batch P", etc as descriptions, name the actual method. Furthermore, Use different colours for the different inoculums. And, give a description of the figure in the caption (what do the letters mean on the error bars, reiterate the abbreviations, make statement about reproducibility, etc.).

Ditto for Figure 2, which I find very confusing.

Ditto for Figures 3 and 4 and 5 and 6 and 7. Overall, tidy up the figure and captions.

The Figures and their captions have been modified according to your suggestions.

Consider whether the data from some of your graphs are better presented in a table.

We have tried to present the information collected in the figures in table format, but the tables obtained were much more complex and less visual.

We hope the modifications incorporated in the figures will make their interpretation easier.

Try not to refer to "Batch S", "Batch P" or "Batch C". I know that is what you might have called them during the experiments, but - keep it simple, call them what they are.

We see your point. Throughout the entire text, we have changed the denomination of the batches. Thus, “Batch C” has be renamed as the "Batch Control", “Batch P” as the "Batch pied de cuve" and “Batch S” as the "Batch ADY".

Round 2

Reviewer 1 Report

Accepted as its revised manuscript.

Accepted as its revised manuscript.

Reviewer 2 Report

The authors have taken all pervious comments into account and addressed them satisfactorily.